# Cross-Generational Effects of Heat Stress on Fitness and *Wolbachia* Density in *Aedes aegypti* Mosquitoes

**DOI:** 10.3390/tropicalmed4010013

**Published:** 2019-01-13

**Authors:** Isabelle Jia-Hui Foo, Ary A. Hoffmann, Perran A. Ross

**Affiliations:** 1Department of Microbiology and Immunology, The Peter Doherty Institute for Infection and Immunity, The University of Melbourne, Victoria 3000, Australia; ifoo@student.unimelb.edu.au; 2Pest and Environmental Adaptation Research Group, School of BioSciences, Bio21 Institute, The University of Melbourne, Victoria 3052, Australia; ary@unimelb.edu.au

**Keywords:** *Aedes aegypti*, *Wolbachia*, heat stress, fitness, starvation tolerance, cross-generational effects

## Abstract

*Aedes aegypti* mosquitoes infected with *Wolbachia* symbionts are now being released into the field to control the spread of pathogenic human arboviruses. *Wolbachia* can spread throughout vector populations by inducing cytoplasmic incompatibility and can reduce disease transmission by interfering with virus replication. The success of this strategy depends on the effects of *Wolbachia* on mosquito fitness and the stability of *Wolbachia* infections across generations. *Wolbachia* infections are vulnerable to heat stress, and sustained periods of hot weather in the field may influence their utility as disease control agents, particularly if temperature effects persist across generations. To investigate the cross-generational effects of heat stress on *Wolbachia* density and mosquito fitness, we subjected *Ae. aegypti* with two different *Wolbachia* infection types (*w*Mel, *w*AlbB) and uninfected controls to cyclical heat stress during larval development over two generations. We then tested adult starvation tolerance and wing length as measures of fitness and measured the density of *w*Mel in adults. Both heat stress and *Wolbachia* infection reduced adult starvation tolerance. *w*Mel *Wolbachia* density in female offspring was lower when mothers experienced heat stress, but male *Wolbachia* density did not depend on the rearing temperature of the previous generation. We also found cross-generational effects of heat stress on female starvation tolerance, but there was no cross-generational effect on wing length. Fitness costs of *Wolbachia* infections and cross-generational effects of heat stress on *Wolbachia* density may reduce the ability of *Wolbachia* to invade populations and control arbovirus transmission under specific environmental conditions.

## 1. Introduction

Viral diseases such as dengue are on the rise. This is due to a suite of factors including shifting geographical distributions of vectors and human mobility around the world, exposing populations to new environmental sources of infectious agents [1]. Dengue is an increasing global threat and is one of the most significant arboviruses. With an increasing geographic range of dengue transmission, it is estimated that about 4 billion people are at risk of dengue infection in 128 countries [2].

Belonging to the Flaviviridae family, dengue virus consists of four different serotypes, namely DENV 1–4 [3]. Primary infection with dengue is usually self-limiting and confers life-long immunity against that serotype. However, a second infection with a different serotype from the first infection increases the risk of severe disease such as dengue shock syndrome and dengue haemorrhagic fever. This is possibly caused by antibody-dependent enhancement of virus infection, whereby cross-reactive antibodies will bind to but not neutralize the virus, worsening the condition of the infection [4]. Today, there are no available anti-viral or effective vaccines to target this disease. 

The primary vector of dengue is *Aedes aegypti*, which is a mosquito that lives in tropical and sub-tropical regions of the world [5]. *Aedes aegypti* has evolved to live in urban environments in close proximity to humans. Female *Ae. aegypti* has a strong preference for feeding on humans, mainly during the daytime, and can enter houses to feed. Once infected with the virus, the mosquito is a carrier for life [6]. *Ae. aegypti* has also evolved to feed on blood from more than one person during a feeding period [7]. These behaviors of *Ae. aegypti* allow efficient transmission of dengue and other pathogenic arboviruses. Hence, reducing or modifying the population of *Ae. aegypti* in nature is essential to reducing the spread of arboviruses such as dengue. 

Conventional approaches to limiting dengue transmission rely on controlling *Ae. aegypti* populations through trapping, removing stagnant water around homes, using mosquito nets and applying insect repellent. Authorities have also implemented routine fogging with adulticides during dengue outbreaks [8]. However, chemical vector control is becoming increasingly ineffective as *Ae. aegypti* has now developed resistance to multiple insecticides in many parts of the world [9]. 

Harnessing *Wolbachia*–mosquito symbiosis is an alternative approach to disease control that could potentially reduce the global burden of dengue and other mosquito-borne diseases [10]. *Wolbachia* are gram-negative bacteria found in many insect species, including mosquitoes, and can interfere with RNA virus replication by competing for cellular components needed for replication, particularly lipids [11]. When transferred from other insect species to *Ae. aegypti* through embryonic microinjection, some *Wolbachia* strains including *w*Mel [12], which is found naturally in *Drosophila melanogaster*, and *w*AlbB [13], which occurs naturally in *Aedes albopictus*, limit the capacity for the mosquitoes to transmit viruses. In *Ae. aegypti*, defective cholesterol and cellular trafficking in *Wolbachia*-infected cells limits viral replication, while restoring cholesterol homeostasis recovers dengue replication in mosquito cells [14].

*Wolbachia* are transmitted maternally and often modify insect reproduction to enhance the production of infected female hosts, facilitating its spread throughout natural populations [10]. Several *Wolbachia* strains that have been transferred to *Ae. aegypti* cause cytoplasmic incompatibility, which results in sperm and eggs being unable to form viable offspring when an uninfected female mates with an infected male [12,15,16]. However, *Wolbachia*-infected females can successfully produce viable and *Wolbachia*-infected offspring with both infected and uninfected males. Cytoplasmic incompatibility can be exploited to introduce *Wolbachia* infections into mosquito populations, transforming natural populations with mosquitoes that are less capable of transmitting arboviruses. Large-scale releases of *Wolbachia*-infected mosquitoes have been carried out in Australia, Malaysia, Vietnam, Brazil and other countries in the tropics with this approach, in an attempt to reduce the burden of dengue [17].

In the field, *Ae. aegypti* larvae experience extreme diurnal fluctuating temperatures, especially in small containers of water that are exposed to direct sunlight or are made of good heat conductors, such as metal [18]. *Ae. aegypti* infected with the *w*Mel strain have greatly reduced *Wolbachia* density when reared at maximum daily temperatures of 37 °C [19,20], similar to temperatures experienced in Cairns, Australia, during the wet season in some breeding sites [18]. Since *Wolbachia* density tends to be positively associated with virus blockage [21,22], heat stress could limit the ability of *Wolbachia*-infected mosquitoes to reduce virus transmission. Heat stress can also reduce the intensity of cytoplasmic incompatibility and fidelity of maternal transmission, potentially impairing the ability of *w*Mel to invade natural populations [20]. However, it is not clear if these effects are transient or if the effects of heat stress persist across generations.

In this study, we looked at the cross-generational effects of heat stress on *Ae. aegypti* fitness and *Wolbachia* density over two generations. The effects on fitness were examined in terms of wing length, which is an indicator of fecundity, and adult starvation tolerance, which is an indicator of nutritional reserves and a trait which has not previously been evaluated in *Wolbachia*-infected mosquitoes. By understanding the cross-generational effects of heat stress, we aim to produce knowledge that can aid the use of *Wolbachia* as an agent for arbovirus control.

## 2. Materials and Methods

### 2.1. Ethics Statement

Blood feeding of female mosquitoes on human volunteers for this research was approved by the University of Melbourne Human Ethics Committee (approval 0723847). All adult subjects provided informed written consent (no children were involved). 

### 2.2. Mosquito Strains and Colony Maintenance

*Ae. aegypti* mosquitoes infected with *w*Mel were collected from Cairns, Queensland, Australia in 2013, in areas where the *w*Mel infection had established [17], and uninfected mosquitoes were collected in 2016 from outside the release area. *Ae. aegypti* infected with *w*AlbB were generated previously by transferring *Wolbachia* from *Ae. albopictus* [16], followed by introgression into an Australian background [23]. *Wolbachia*-infected females were crossed to uninfected males regularly to maintain all colonies on a similar genetic background. All populations were maintained in the laboratory as described by Ross et al. [24]. 

### 2.3. Mosquito Rearing at Constant and Cyclical Temperatures

We investigated the effect of heat stress during larval development on mosquito fitness and *Wolbachia* density in *Ae. aegypti* across two generations. In the first generation, eggs of uninfected, *w*Mel-infected and *w*AlbB-infected mosquitoes were hatched concurrently in 3 L trays of reverse osmosis (RO) water at 26 °C and provided with one 300 mg tablet of tropical fish food (TetraMin, Tetra, Melle, Germany). Hatching trays were kept in a controlled-temperature room at a constant temperature of 26 °C and a 12/12 h light/dark cycle. 

Twenty-four hours after hatching, 200 larvae were added to plastic containers filled with 500 mL of RO water to control the larval density. Larvae were either reared at a constant temperature of 26 °C or subjected to a diurnal cyclical temperature of 26–37 °C (Table 1) in incubators (PG50 Plant Growth Chambers, Labec Laboratory Equipment, Marrickville, NSW, Australia). The temperature cycle of 26–37 °C was consistent with previous studies [19,20] and was chosen to reflect temperatures experienced in breeding sites in Cairns during the wet season [18]. Four replicate trays of larvae were reared for each *Wolbachia* infection type and temperature. Temperature loggers (Thermochron; 1-Wire, iButton.com, Dallas Semiconductors, Sunnyvale, CA, USA) in zip-lock bags were placed in six containers at random to monitor temperature in the incubator (Figure A1). The position of the containers was randomly shuffled every day to account for location-dependent temperature effects.

The larvae were reared to the pupal stage by providing fish food tablets ad libitum and then returned to a constant temperature of 26 °C for the adult and egg stages. Emerging adults were collected at random and tested for their starvation tolerance (see below), while the remainder were maintained for experiments on the second generation. One week after adults emerged, the females were blood-fed on a single human volunteer, their eggs were collected and conditioned, then hatched four days after collection. Larvae from each infection type and rearing temperature in the first generation were reared at either 26 °C or 26–37 °C in the second generation, for a total of four temperature treatments for each infection type (Table 1). When adults from the second generation emerged, 20 males and 20 females from each treatment were selected at random and stored in absolute ethanol for wing length measurements and *Wolbachia* density quantification. The remaining adults were used in the starvation tolerance experiment.

### 2.4. Adult Starvation Tolerance

We investigated the effect of heat stress during larval development on adult starvation tolerance in the first and second generation. Starvation resistance has been extensively studied in other Diptera including *Drosophila* where this trait interacts with other forms of stress resistance and can be affected by carryover effects [25]. Adult mosquitoes were subjected to starvation conditions whereby 25 males and 25 females from each *Wolbachia* infection type and rearing temperature were transferred to 3 L cages and provided with water only. Each treatment was replicated four times. Adult starvation tolerance was determined by counting and removing dead mosquitoes from the cages each day until the last adult died.

For the experiment on the first generation, adult mosquitoes were provided with 10% sucrose for 4 days before being transferred to experimental cages. In experiments on the second generation, adult mosquitoes were not provided with sugar before the experiment and instead were transferred to 3 L cages within one day of emerging. 

### 2.5. Wing Length

To determine the effect of heat stress on body size in the second generation, 15 wings from each sex and treatment were dissected and mounted on glass slides with Hoyer’s solution (dH_2_O/gum arabic/chloral hydrate/glycerin in the ratio 5:3:20:2). Wing length was measured as the distance from the axial notch to the wing tip [26] using NIS-Elements BR (Nikon Instruments, Japan). Each measurement was repeated, and lengths were averaged to produce a final measurement.

### 2.6. Wolbachia Density

*Wolbachia* density for *w*Mel-infected mosquitoes was quantified using a quantitative real-time polymerase chain reaction (RT-qPCR) assay. Genomic DNA was extracted from 12 adult mosquitoes from each treatment, using 150 μL of 5% Chelex 100 resin (Bio-Rad Laboratories, Hercules, CA, USA). *Ae. aegypti*-specific primers involving a region in the *RpS6* gene and *w*Mel-specific primers involving the *w1* marker were then used to quantify *Wolbachia* density relative to the density of mosquito DNA with an RT-qPCR assay outlined by Lee and others [27]. PCR was carried out using a Roche LightCycler 480 system (Roche Applied Science, Indianapolis, IN) to obtain crossing point (Cp) values for these markers for each mosquito. Differences between the Cp of the *Wolbachia* and *Ae. aegypti* markers were transformed by 2^n^ to obtain approximate estimates of *Wolbachia* density.

### 2.7. Statistical Analysis

All data collected were analysed using GraphPad Prism v.7. Log-rank Mantel Cox tests were performed to compare differences in adult starvation tolerance between groups. We used general linear models to investigate the effects of sex, *Wolbachia* infection type and temperature regime on wing length and to compare *Wolbachia* density between temperature regimes. 

## 3. Results

### 3.1. Adult Starvation Survival

#### 3.1.1. Generation 1

We tested the tolerance of uninfected, *w*Mel-infected and *w*AlbB-infected adults to starvation when larvae were reared at 26 °C or 26–37 °C (Figure 1). Adults that were reared at 26 °C had a higher starvation tolerance than those reared at 26–37 °C, with differences being highly significant for both females (Log-rank test: χ^2^ = 9.493, df = 1, *p* = 0.002) and males (χ^2^ = 18.243, df = 1, *p* < 0.001). Differences between males and females were substantial (χ^2^ = 537.901, df = 1, *p* < 0.001), with females outliving males under starvation conditions. *w*Mel-infected (females: χ^2^ = 65.814, df = 1, *p* < 0.001, males: χ^2^ = 8.414, df = 1, *p* = 0.004) and *w*AlbB-infected (females: χ^2^ = 94.552, df = 1, *p* < 0.001, males: χ^2^ = 37.541, df = 1, *p* < 0.001) adults had reduced starvation tolerance relative to uninfected mosquitoes. *w*AlbB-infected adults had lower starvation tolerance than *w*Mel-infected adults for both females (χ^2^ = 5.546, df = 1, *p* = 0.019) and males (χ^2^ = 19.960, df = 1, *p* < 0.001).

#### 3.1.2. Generation 2

We evaluated the cross-generational effects of heat stress on adult starvation tolerance. In contrast to the experiments on the first generation, males had a higher tolerance to starvation than females (Log-rank test: χ^2^ = 154.502, df = 1, *p* < 0.001). This likely reflects differences in methodology; adults were starved immediately after emergence in the second generation, while in the first generation they were fed sucrose first.

Adults reared at 26 °C in the second generation had higher starvation tolerance than adults reared at 26–37 °C (females: χ^2^ = 94.661, df = 1, *p* < 0.001, males: χ^2^ = 56.243, df = 1, *p* < 0.001), consistent with the first experiment. We found cross-generational effects of heat stress on starvation tolerance in females but not in males (Figure 2). Females reared at 26°C in the second generation had higher starvation tolerance when their parents had experienced heat stress (χ^2^ = 49.504, df = 1, *p* < 0.001), but there was no cross-generational effect when second-generation females were reared at 26–37°C (χ^2^ = 3.635, df = 1, *p* = 0.057). There was no cross-generational effect of heat stress in males; starvation tolerance when the second generation was reared at 26 °C (χ^2^ = 1.386, df = 1, *p* = 0.239) or 26–37°C (χ^2^ = 3.373, df = 1, *p* = 0.066) was unaffected by the rearing temperature of the first generation.

When considered across all temperatures, uninfected adults had a higher starvation tolerance than *Wolbachia*-infected adults for both males (*w*Mel: χ^2^ = 40.884, df = 1, *p* < 0.001; *w*AlbB: χ^2^ = 16.743, df = 1, *p* < 0.001) and females (*w*Mel: χ^2^ = 37.821, df = 1, *p* < 0.001; *w*AlbB: χ^2^ = 46.963, df = 1, *p* < 0.001, Figure 3). *w*AlbB-infected males had higher starvation tolerance than *w*Mel-infected males (χ^2^ = 7.227, df = 1, *p* = 0.007), but tolerance did not differ between the two strains for females (χ^2^ = 0.059, df = 1, *p* = 0.809).

### 3.2. Wing Length

We measured wing length from a sample of adults from the second generation as an estimate of body size. Females (mean ± SE = 2.835 ± 0.0155 mm, *n* = 160) were much larger than males (mean ± SE = 2.167 ± 0.0113 mm, *n* = 147, general linear model: F_1,305_ = 1182.338, *p* < 0.001, Figure 4). There was a significant effect of the *Wolbachia* infection type on wing length for males (F_2,135_ = 13.982, *p* < 0.001), with *w*Mel-infected males being smaller than the other two infection types, but there was no effect in females (F_2,148_ = 1.553, *p* = 0.215). We found no cross-generational effects for this trait; heat stress during the first generation had no bearing on wing length in the second generation for both males (F_1,135_ = 0.101, *p* = 0.751) and females (F_1,148_ = 2.090, *p* = 0.150). However, differences related to the rearing temperatures during the second generation were substantial for both males (F_1,135_ = 522.311, *p* < 0.001) and females (F_1,148_ = 416.768, *p* < 0.001), with mosquitoes reared at 26 °C being much larger than mosquitoes reared at 26–37°C (Figure 4).

### 3.3. Wolbachia Density

We measured *Wolbachia* density in *w*Mel-infected adults to see if the rearing temperature in the first generation affected density in the second generation (Figure 5). *Wolbachia* density was higher in adults reared at 26 °C during larval development than in adults reared at 26–37 °C (females: general linear model: F_1,22_ = 176.844, *p* < 0.001, males: F_1,22_ = 100.051, *p* < 0.001). We found cross-generational effects of heat stress on *Wolbachia* density in females but not in males. *Wolbachia* density was lower in females reared at 26°C in the second generation when the first generation was reared under heat stress (F_1,22_ = 44.391, *p* < 0.001). *Wolbachia* density was also lower in females reared at 26–37 °C in the second generation when the first generation was reared under heat stress, but this difference was marginally non-significant (F_1,22_ = 4.140, *p* = 0.054). In contrast to females, there was no cross-generational effect of heat stress on *Wolbachia* density in males for either rearing temperature in the second generation (26°C: F_1,22_ = 0.466, *p* = 0.502, 26–37 °C: F_1,20_ = 1.688, *p* = 0.211). These results indicate that the effects of heat stress on female *Wolbachia* density can accumulate across generations, but *Wolbachia* density can also partially recover in the next generation in the absence of heat stress.

## 4. Discussion

In this study we evaluated the cross-generational effects of heat stress on *Wolbachia* density and fitness in *Wolbachia*-infected and uninfected *Ae. aegypti*. We found that diurnal fluctuating temperatures of 26–37 °C during larval development reduced adult starvation tolerance, wing length and *Wolbachia* density. Heat stress during the first generation influenced *Wolbachia* density and adult starvation tolerance in the following generation, but only in females. We also found that *Wolbachia* infections reduced adult starvation tolerance, adding to the growing list of phenotypic effects influenced by *Wolbachia* infections in *Ae. aegypti* [28].

Heat stress during development reduces *Wolbachia* density and cytoplasmic incompatibility intensity in *w*Mel-infected *Ae. aegypti* [20,29,30], though *Wolbachia* density seems to partially recover when individuals are returned to cooler temperatures [19]. Few studies have measured the effects of temperature on *Wolbachia* infections across generations, except in cases where high temperatures were being used to deliberately cure *Wolbachia* infections [31], though these tended to measure infection frequencies rather than density [32,33]. In the parasitoid wasp *Leptopilina heterotoma*, *Wolbachia* density in daughters was not influenced by the rearing temperature of the mother (20 or 26 °C) [34], but the effects of temperature on *Wolbachia* density across generations have not been tested previously in *Ae. aegypti*.

In this study, we found that *Wolbachia* density partially recovered in female offspring of parents that experienced heat stress. In contrast, there was no effect of heat stress during the first generation on *Wolbachia* density in the second generation for males. This result was unexpected given that *Wolbachia* density responses to high temperatures are usually similar between males and females in *Ae. aegypti* [20,29]. Our results suggest that the effects of heat stress on *Wolbachia* density can accumulate across generations, at least in females. Long periods of hot weather (across multiple mosquito generations) could lead to a continuous decline in *Wolbachia* density, while short periods within a single generation may affect density in the following generation. Given that *Wolbachia* density is positively associated with virus blockage [21,22,35,36] and cytoplasmic incompatibility [37,38] in other systems, there may also be cross-generational effects of heat stress on these phenotypes in *Ae. aegypti*. However, since *Wolbachia* density partially recovered in the next generation in the absence of heat stress, it is unlikely that these effects would persist for more than a couple of generations under cooler conditions. 

A large range of fitness effects have now been identified for *Wolbachia* infections in *Ae. aegypti*, with most research focusing on the *w*MelPop and *w*Mel strains which have both been deployed into the field in disease control programs [17,39,40,41]. *Wolbachia* infections in *Ae. aegypti* tend to reduce fitness, particularly in terms of fertility [42], adult lifespan [15,23,43] and quiescent egg viability [44,45]. Despite much research on fitness effects of *Wolbachia* infections, there is relatively little research on the effects of *Wolbachia* infections on starvation tolerance. Previous studies in adult *Drosophila* found no effect of *Wolbachia* on this trait [46,47], but the *w*Mel, *w*AlbB and *w*MelPop infections in *Ae. aegypti* reduce the survival of larvae without food, with the severity of the effect depending on the strain [48]. Johnson et al. [49] evaluated the survival of wild-type and *w*Mel-infected male *Ae. aegypti* adults after the removal of honey solution and found that wild-type males survived slightly longer, though the two infection types were not compared statistically.

Here, we show that the *Wolbachia* infections *w*Mel and *w*AlbB reduced adult starvation tolerance in *Ae. aegypti*, with no consistent differences between the two strains when considering both experiments. *Wolbachia* infections in *Ae. aegypti* can increase adult metabolic rate [50] and compete with their host for resources [51], which may explain the costs to starvation tolerance seen here. While the effects of *Wolbachia* on adult starvation tolerance are relatively subtle, fitness may be reduced under certain conditions in the field where nutrition is not readily available. Since female *Ae. aegypti* rarely feed on sugar [52], *Wolbachia* infections may shorten lifespan if females are not able to feed on blood regularly. *Wolbachia* may also reduce the opportunity for males to reproduce in the absence of sugar, which is their only source of nutrition. 

## 5. Conclusions

In summary, we demonstrated that diurnal fluctuating temperatures of 26–37 °C during *Ae. aegypti* larval development reduced *w*Mel *Wolbachia* density, with effects persisting into the next generation for females but not males. These findings have implications for the stability of *Wolbachia* infections across generations in tropical environments. The costs of *Wolbachia* infections to adult starvation tolerance observed here add to an increasing array of traits that are adversely affected by *Wolbachia*, which together will reduce the potential for *Wolbachia* to invade natural populations of *Ae. aegypti*.

## Figures and Tables

**Figure 1 tropicalmed-04-00013-f001:**
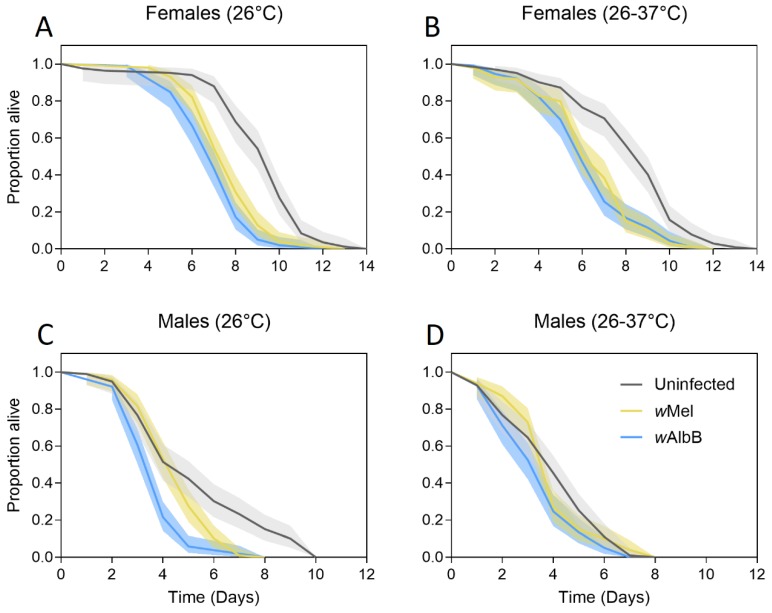
Adult starvation tolerance of female (**A**,**B**) and male (**C**,**D**) *Ae. aegypti* that were reared at either 26 °C (**A**,**C**) or 26–37 °C (**B**,**D**). Mosquitoes were either uninfected (gray lines) or infected with the *w*Mel *Wolbachia* strain (yellow lines) or the *w*AlbB *Wolbachia* strain (blue lines). Solid lines represent the proportion of adults alive each day, while shaded areas are 95% confidence intervals.

**Figure 2 tropicalmed-04-00013-f002:**
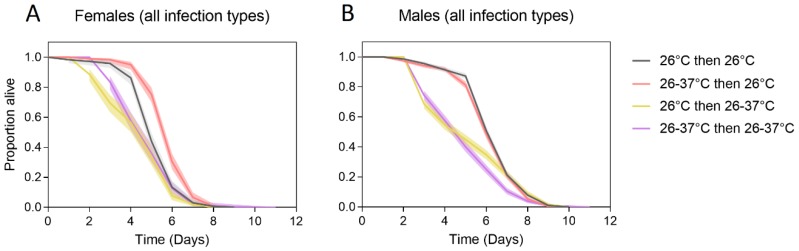
Adult starvation tolerance of female (**A**) and male (**B**) *Ae. aegypti* that were reared under different temperature regimes across two generations. Mosquitoes were reared at 26 °C in both generations (gray lines), 26–37 °C in both generations (purple lines), 26–37 °C in the first generation and 26 °C in the second generation (red lines) or 26 °C in the first generation and 26–37 °C in the second generation (yellow lines). Data for all *Wolbachia* infection types were pooled in these comparisons. Solid lines represent the proportion of adults alive each day, while shaded areas are 95% confidence intervals.

**Figure 3 tropicalmed-04-00013-f003:**
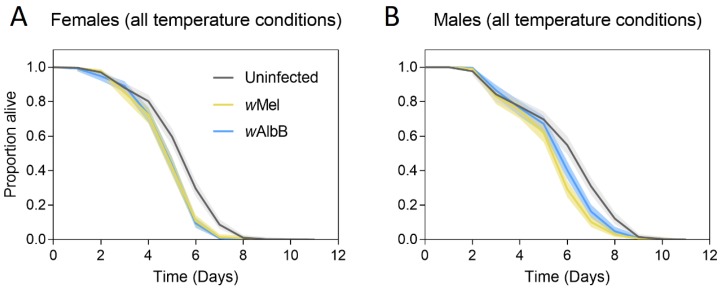
Adult starvation tolerance of female (**A**) and male (**B**) *Ae. aegypti* with different *Wolbachia* infection types in the second generation. Mosquitoes were either uninfected (gray lines) or infected with the *w*Mel *Wolbachia* strain (yellow lines) or the *w*AlbB *Wolbachia* strain (blue lines). Data for all rearing temperature regimes were pooled in these comparisons. Solid lines represent the proportion of adults alive each day, while shaded areas are 95% confidence intervals.

**Figure 4 tropicalmed-04-00013-f004:**
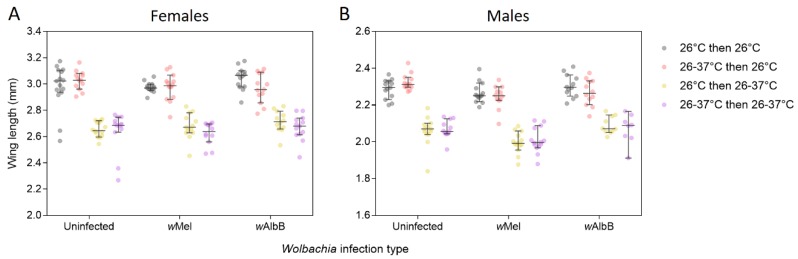
Wing length of female (**A**) and male (**B**) *Ae. aegypti* from three *Wolbachia* infection types (uninfected, *w*Mel-infected or *w*AlbB-infected) that were reared under different temperature regimes across two generations. Mosquitoes were reared at 26 °C in both generations (gray dots), 26–37 °C in both generations (purple dots), 26–37 °C in the first generation and 26 °C in the second generation (red dots) or 26 °C in the first generation and 26–37 °C in the second generation (yellow dots). Error bars are medians and 95% confidence intervals.

**Figure 5 tropicalmed-04-00013-f005:**
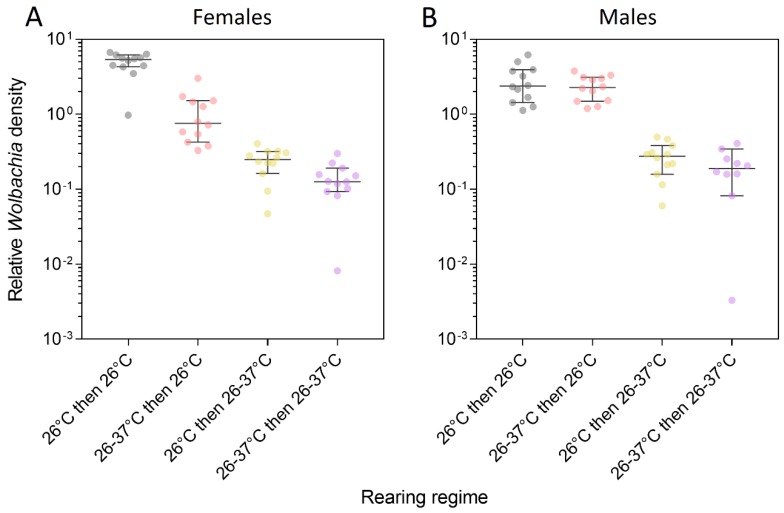
Relative *Wolbachia* density of *w*Mel-infected male and female *Ae. aegypti* that were reared under different temperature regimes across two generations. Mosquitoes were reared at 26 °C in both generations (gray dots), 26–37 °C in both generations (purple dots), 26–37 °C in the first generation and 26 °C in the second generation (red dots) or 26 °C in the first generation and 26–37 °C in the second generation (yellow dots). Error bars are medians and 95% confidence intervals.

**Table 1 tropicalmed-04-00013-t001:** Experimental design. *Aedes aegypti* mosquito larvae for all three infection types (uninfected, *w*Mel, and *w*AlbB) were reared at either 26 °C or 26–37 °C in the first generation. Offspring of parents from each treatment were then reared at either 26 °C or 26–37 °C.

Infection Type	Generation	
	1	2
Uninfected	26 °C	26 °C
	26 °C	26–37 °C
	26–37 °C	26 °C
	26–37 °C	26–37 °C
*w*Mel	26 °C	26 °C
	26 °C	26–37 °C
	26–37 °C	26 °C
	26–37 °C	26–37 °C
*w*AlbB	26 °C	26 °C
	26 °C	26–37 °C
	26–37 °C	26 °C
	26–37 °C	26–37 °C

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
