# Peer review of "Cross-Generational Effects of Heat Stress on Fitness and Wolbachia Density in Aedes aegypti Mosquitoes"

_tropicalmed, 2019, doi:10.3390/tropicalmed4010013_

Round 1

Reviewer 1 Report

Wolbachia infected mosquitoes are being released in the field for controlling dengue transmission.  In order to better design the release programs, it is essential to precisely characterize the fitness cost of Wolbachia infection. Temperatures in nature can be suboptimal for both Wolbachia and mosquito hosts, thus studies on the effect of temperature on Wolbachia-host interaction are important.  The current study investigates the effects of diurnal temperature fluctuation as a heat stress (26-37C) during larval stage on the fitness of adult mosquitoes (starvation tolerance and size) and Wolbachia density in adult mosquitoes within the same generation as well as in the next generation using an uninfected and two Wolbachia infected Aedes aegypti lines.  Authors demonstrate both heat stress and Wolbachia infection could negatively impact the starvation tolerance of adult mosquitoes, and there were cross-generational effects on starvation tolerance and Wolbachia density.

Overall, the data presentation was clear, and discussions/conclusions were appropriate.  A minor thing to address might be justifying a bit more regarding why authors chose to look at the adult starvation tolerance since little is known how much adult mosquitoes experience starvation in the field.  Additionally, it was not quite clear why fecundity, cytoplasmic incompatibility, or maternal inheritance of the Wolbachia infections were not measured directly because these are the important parameters that potentially determine the success of release studies. Lastly, a bit of clarification might be helpful regarding the methods in lines 163-164.  

Author Response

A minor thing to address might be justifying a bit more regarding why authors chose to look at the adult starvation tolerance since little is known how much adult mosquitoes experience starvation in the field. Additionally, it was not quite clear why fecundity, cytoplasmic incompatibility, or maternal inheritance of the Wolbachia infections were not measured directly because these are the important parameters that potentially determine the success of release studies.

We chose adult starvation tolerance as an indicator of nutritional reserves, and we expected this to differ between mosquitoes reared at different temperatures. Adult starvation resistance is also a trait that has been commonly studied in other Diptera including Drosophila and considered for carry over effects. We would have preferred to test for other traits but were constrained by limited resources. Some wording changes have been made, including a short mention of other studies on starvation resistance (Hoffmann and Harshman 1999, added as reference 25 in text).

Lastly, a bit of clarification might be helpful regarding the methods in lines 163-164.

We’ve reworded this a bit. “Aedes aegypti-specific primers involving a region in the RpS6 gene and wMel-specific primers involving the w1 marker were then used to quantify Wolbachia density relative to the density of mosquito DNA with an RT-qPCR assay outlined by Lee and others [26]. PCR was carried out using a Roche LightCycler 480 system (Roche Applied Science, Indianapolis, IN) to obtain crossing point (Cp) values for these markers for each mosquito. Differences between the Cp of the Wolbachia and Ae. aegypti markers were transformed by 2n to obtain approximate estimates of Wolbachia density.”

Reviewer 2 Report

Authors investigate cross-generational effects of larval heat stress on adult fitness and Wolbachia density in Ae. aegypti. Two two Wolbachia  infection (wMel, wAlbB) were compared to uninfected controls. Adult fitness effects were tested for starvation tolerance, wing length.  Authors found that heat stress reduced adult starvation tolerance and female wMel Wolbachia density in offspring was lower when mothers experienced heat stress. However male Wolbachia density did not depend on the rearing temperature of the previous generation. There appeared to be cross-  generational effects of heat stress on female starvation tolerance but no effect on wing length.

Major comments

The MS is well written and clearly presented, figures appear sufficient to detail results and referencing is satisfactory. My only real gripe is the temperature fluctuation selection of 37-26 degree – please justify selection.

Minor comment

The acronym/abbreviation of wMel/wAlb usually does not have a space between the “w” and abbreviation of originating arthropod species the Wolbachia came from.

Author Response

Major comments

The MS is well written and clearly presented, figures appear sufficient to detail results and referencing is satisfactory. My only real gripe is the temperature fluctuation selection of 37-26 degree – please justify selection.

The temperature cycle was chosen for the sake of consistency with previous studies which found an effect on Wolbachia density at this temperature. This temperature range is also similar to maximum daily temperatures experienced in some larval habitats in Cairns, Australia during the wet season. We have clarified this in the methods section.

Minor comment

The acronym/abbreviation of wMel/wAlb usually does not have a space between the “w” and abbreviation of originating arthropod species the Wolbachia came from.

We double checked with the search function and found no cases of this occurring in the text. The Wolbachia strains are always written as wMel and wAlbB without spaces.

Reviewer 3 Report

I have reviewed the manuscript titled “Cross-generational effects of heat stress on fitness and Wolbachia density in Aedes aegypti mosquitoes” by Jia-Hui Foo et al. The manuscript reports on laboratory experiments designed to assess the impact of heat stress on immature mosquitoes infected with Wolbachia as measured by starvation tolerance and wing length across two generations. The paper is well written and the research outcomes are of importance for those working in the field of mosquito research and operations to implement mosquito management strategies that incorporate the release of laboratory reared mosquitoes infected with Wolbachia.

I have no major comments regarding the content or presentation of the research. However, there are a small number of minor suggestions for the authors to address in their Materials and Methods or Discussion sections that could improve the manuscript.

While the authors state in the Introduction that previous studies demonstrated that 37oC greatly reduces Wolbachia density in Aedes aegypti, it would be useful to provide specific reasons why the temperature range 26-37oC was used in the experiments. Does this temperature range represent specific geographical locations or is it informed by other research? It would also be useful if the authors include some comment in the Discussion as to how representative the selected temperature range is in “real world” scenarios. I appreciate that this may be beyond the scope of the manuscript but it may be worth including a brief comment on how likely mosquitoes are to be associated with exposed water-holding containers that are likely to reach such high, and sustained, temperatures in the field.

Wing length is a commonly used measure of larval nutrition and subsequent body size. Studies have indicated that wing length/body size can be influenced by larval development times, those development times also influenced by provision of food and temperature. It would be worth noting any observed differences in larval development times of the experimental cohorts. With an overall mean temperature well above 26oC, I suspect that larval development was faster in the treatment replicates than control. This is likely a contributing factor leading to the finding that mosquitoes reared at 26oC were much larger than those reared at 26-37oC. Similarly, is there any evidence to suggest that Wolbachia infection rates in adult mosquitoes are influenced by larval development rates?

Author Response

While the authors state in the Introduction that previous studies demonstrated that 37oC greatly reduces Wolbachia density in Aedes aegypti, it would be useful to provide specific reasons why the temperature range 26-37oC was used in the experiments. Does this temperature range represent specific geographical locations or is it informed by other research? It would also be useful if the authors include some comment in the Discussion as to how representative the selected temperature range is in “real world” scenarios. I appreciate that this may be beyond the scope of the manuscript but it may be worth including a brief comment on how likely mosquitoes are to be associated with exposed water-holding containers that are likely to reach such high, and sustained, temperatures in the field.

We now state in the methods that this temperature cycle was chosen based on temperatures that are experienced by larvae in the field in Cairns, Australia (where the first releases of Wolbachia-infected Ae. aegypti took place), and we now also clarify this in the introduction. There are few container surveys that measure temperature so it is difficult to say how commonly these temperatures will be experienced, but the proportion of breeding sites that will adversely affect Wolbachia is likely to be highly variable and differ between release locations. We discuss this issue in an upcoming paper which looks at temperature effects under more natural conditions.

Wing length is a commonly used measure of larval nutrition and subsequent body size. Studies have indicated that wing length/body size can be influenced by larval development times, those development times also influenced by provision of food and temperature. It would be worth noting any observed differences in larval development times of the experimental cohorts. With an overall mean temperature well above 26oC, I suspect that larval development was faster in the treatment replicates than control. This is likely a contributing factor leading to the finding that mosquitoes reared at 26oC were much larger than those reared at 26-37oC. Similarly, is there any evidence to suggest that Wolbachia infection rates in adult mosquitoes are influenced by larval development rates?

Unfortunately we didn’t test larval development time in this experiment since we would have needed greater replication, but from previous studies we find that larvae reared at 26-37 are slightly quicker to develop, as you suspect (and we agree that this is likely a contributing factor). We found that larval development time can affect Wolbachia density in one of our previous studies (https://www.ncbi.nlm.nih.gov/pubmed/24732463), where larvae that are slower to develop have a higher density. However, there is also clearly a direct effect of temperature that occurs independently from development time (see figure S4 in https://www.mdpi.com/2075-4450/9/3/78/htm).